# Development of a Low-Cost, Wireless Smart Thermostat for Isothermal DNA Amplification in Lab-On-A-Chip Devices

**DOI:** 10.3390/mi10070437

**Published:** 2019-06-30

**Authors:** Tamas Pardy, Henri Sink, Ants Koel, Toomas Rang

**Affiliations:** Thomas Johann Seebeck Department of Electronics, Tallinn University of Technology, 19086 Tallinn, Estonia

**Keywords:** Lab-on-a-Chip (LoC), finite element modelling, resistive heating, Point-of-Care (PoC), temperature control, computer-aided design, microfluidics, isothermal nucleic acid amplification tests, noninstrumented nucleic acid amplification tests (NINAAT), nucleic acid amplification tests (NAAT)

## Abstract

Nucleic acid amplification tests (NAAT) are widely used for the detection of living organisms, recently applied in Lab-on-a-Chip (LoC) devices to make portable DNA analysis platforms. While portable LoC-NAAT can provide definitive test results on the spot, it requires specialized temperature control equipment. This work focuses on delivering a generalized low-cost, wireless smart thermostat for isothermal NAAT protocols in 2 cm × 3 cm LoC cartridges. We report on the design, prototyping, and evaluation results of our smart thermostat. The thermostat was evaluated by experimental and simulated thermal analysis using 3D printed LoC cartridges, in order to verify its applicability to various isothermal NAAT protocols. Furthermore, it was tested at the boundaries of its operating ambient temperature range as well as its battery life was evaluated. The prototype thermostat was proven functional in 20–30 °C ambient range, capable of maintaining the required reaction temperature of 12 isothermal NAAT protocols with 0.7 °C steady-state error in the worst case.

## 1. Introduction

Nucleic acid amplification test (NAAT) protocols are currently the gold standard in detecting living organisms on a cellular level, PCR (polymerase chain reaction) being the most widely used protocol [1,2]. A significant amount of effort goes into transferring NAATs into portable devices, in particular Lab-on-a-Chip (LoC) devices, for analytical or diagnostic field applications [3,4,5,6]. Isothermal NAAT protocols are particularly suitable for this format as, unlike PCR, they only require a single temperature target range to be maintained; therefore, they are easier to implement on compact, low cost hardware [7,8,9,10]. To facilitate the large-scale application of portable isothermal DNA amplification LoC devices, compact, low-cost, user-friendly electronics are needed, such as portable micropumps and heaters with built-in thermostats. 

Heating elements used in LoC are typically electrical, with two main categories: external Peltier [11,12,13,14] and integrated resistive [15,16,17] or micro-Peltier [18,19,20]. All of these are regulated by thermostats that are typically external to the LoC device. Self-regulating resistive heating elements are an interesting novel solution to integrated temperature control as they require no thermostat [21,22,23,24,25]. However, due to the open-loop nature of the PTCR (positive temperature coefficient of resistance) effect used for control, they require extensive optimization to the particular NAAT protocol and ambient temperature conditions they are used in [21]. Noncontact electrical heating solutions include induction and microwave heating [26], but are impractical for portable applications. Lastly, nonelectrical solutions exist as well. Exothermic chemical reactions regulated by engineered phase-change materials offer extremely cheap (~1 EUR) integrated heating [27,28,29,30], but their temperature control is challenging and they all require extensive optimization to their targeted operating conditions. In summary, general applicability to isothermal NAAT is only possible with the closed-loop control (typically using the well-known proportional-integral-derivative, aka. PID algorithm) that a thermostat can provide, and preferably using a resistive heating element.

Reported state of the art LoC NAAT applications typically focus on the PCR protocol and use external/integrated thermal cyclers [31,32,33,34]. For isothermal NAAT, it is possible to use a laboratory hot plate [35], a thermostat built for microfluidic chips [36] or even thermal cyclers programmed to hold at a specific temperature [37]. All methods employ closed-loop control and electrical heating (Peltier elements or resistive heating). Portable thermal regulators/cyclers for LoC NAAT fall into three categories regarding cost and product maturity: (1) prototypes for 100–500 € [38,39,40,41], (2) early market entries by start-up companies for 500–2500 € [42,43], and (3) mature commercial products from established brands for over 2500 € [26,36,44]. Table 1 shows a comparison of temperature control solutions in the literature with respect to features and cost. 

In previous works, we demonstrated the integration of commercially available electrical heating elements into isothermal LoC NAAT devices [46,47,48,49,50]. Furthermore, we demonstrated the application of a self-regulating PTCR polymer-resin heating element in a LoC NAAT device, including thermal analysis [21,51] and proof-of-concept LAMP tests [22]. Previously we focused exclusively on low-cost, integrated heating solutions for use specifically in single-use LoC devices and primarily on the LAMP assay. As the first step in developing a low-cost LoC platform for a wide variety of isothermal DNA amplification protocols, in this work, we introduce a novel low-cost, wireless smart thermostat. The device is battery-powered and offers remote control via Bluetooth. Contrary to our previous works, this work demonstrates an external thermostat applicable to LoC cartridges of specific dimensions, irrespective of microchannel geometry. The focus is on finding the best balance between low cost, user-friendliness, and performance. This paper reports on the thermostat design, as well as its validation with experimental and simulated thermal analysis at temperature targets required by a wide array of isothermal DNA amplification assays. Experimental thermal analysis is performed on 3D printed microfluidic cartridges with the standard 0.05 mL amplification volume, both by thermistor probes embedded in the channel and infrared thermometry to observe spatial distribution of heat. Simulation via finite element modelling is used to verify that the reaction volume is in range in steady state at each temperature target, as well as to define the best steady-state error achievable by the system. The thermal evaluation methodology used in this paper was demonstrated in our previous works [21,52]. 

## 2. Materials and Methods

### 2.1. Thermostat Prototype Design 

The thermostat prototype (Figure 1) consisted of the electronics assembly, heater, microfluidic chip, battery, and the enclosure. The electronics assembly was constructed from a set of stacked circuit boards: an electronics interface board, a microcontroller board, and a user interface board. The enclosure included user input/output interfaces. Key design aspects were simplicity (cheap fabrication and materials), scalability, and user-friendliness. In this section, we discuss the electronics assembly (including the battery and user interfaces) and the enclosure. The heater is detailed in Section 2.2.

The core component of the electronics assembly was the microcontroller board based on the ESP32 SoC (System-on-Chip) [53]. We used the TTGO MINI 32 V2.0 development board (WeMos Inc., Guangdong, China), which was chosen due to its low cost, small footprint, integrated battery management, integrated Bluetooth 4.2/BLE (Bluetooth Low Energy) communication capabilities, and Arduino compatibility. The SoC board was connected to the electronics interface board, which was connected to the user interface board, the battery and the heater. The input current of the heater was regulated through an IRLZ44N (Infineon Technologies AG, Neubiberg, Germany) nMOSFET (metal-oxide-semiconductor field-effect transistor), controlled in turn by a GPIO (general purpose input-output) pin of the ESP32 using PWM (pulse width modulation). Additionally, the interface board had a TI ADS1115 (Texas Instruments, Dallas, Texas, USA) 16-bit low-power ADC (analog to digital converter) for temperature measurement (from the thermistor on the heater board), heater input current and battery voltage output measurement. The battery was a 3.7 V, 1200 mAh Lithium-Ion battery. The device was designed to have two user input/output interfaces: a built-in OLED (organic light-emitting diode) display with three pushbuttons on the user interface board, and a Bluetooth serial interface for wireless control. The OLED display was controlled via the I2C interface of the ESP32. Two of the pushbuttons navigated up/down in the user menu and a third selected menu items. The Bluetooth serial interface was implemented using the BluetoothSerial Arduino library for the ESP32 by Espressif [54], which could offer up to 10 m communication range in an office environment [55]. A smartphone Bluetooth client was created in MIT App Inventor 2 for Android for configuring the thermostat remotely [56]. 

The enclosure was 3D printed by an FDM (fused deposition modelling) 3D printer (Ultimaker 2+, Ultimaker B.V., Geldermalsen, Netherlands) from ABS (acrylonitrile butadiene styrene) plastic. The prototype enclosure was 12 cm × 8 cm × 5 cm in total. One side of the enclosure was for user input-output, the other for the microfluidic chip interface. On the chip interface side, the heater was fastened to a slot perpendicular to the slot for the microfluidic chip. The chip was clamped down by a thick plastic cover that also acted as thermal insulation. The plastic cover had holes for the silicone tubing connected to the microfluidic chip. On the user interface side, above the electronics assembly, the enclosure had holes for the pushbuttons and a slot for the OLED display. The vertical wall next to the user interface had a slot for connecting cables to the electronics inside.

### 2.2. Heating Element, Regulation, and Thermal Interface 

In this work, the decision was made to use printed circuit boards (PCBs) for heating due to their low cost (~0.03 €/pc), easy integration, and low thermal conductivity (0.3 W/(m·K)). This solution is similar to etched foil or patterned thin film resistive heaters, but instead of commonly used resistive materials, copper tracks are used for heating. Due to the low resistivity of copper, this results in a low-power resistive heater. Furthermore, even the cheapest PCBs have excellent height uniformity, eliminating the need for additional heat spreaders or thermal compounds on the thermal interface. The prototype we designed had a copper heating track (~0.036 mm layer thickness, 4 cm^2^ area) and copper fills in between to spread heat evenly, covered by solder mask for electrical insulation. In the middle of the heater there was a copper fill and vias to the backside where the thermistor was located (position TH1 in Figure 1). This was to ensure good heat transfer to the thermistor. A 10K NTC thermistor (NCP21XV103J03RA, Murata Manufacturing Co., Ltd., Nagaokakyo, Kyoto, Japan) was used for providing temperature feedback for controlling the output of the heater. The heater and the thermistor were connected by a four-pin header (J1 in Figure 1). Prototypes were ordered from JLCPCB (Shenzhen JIALICHUANG Electronic Technology Development Co., Ltd., Shenzhen, China). 

The heater’s current input was regulated by an nMOSFET, in turn controlled from one of the GPIO interfaces of the ESP32 (see Section 2.1). Heater power control was realized by PWM (pulse-width modulation) control of the FET’s output. The control algorithm was based on a PID (proportional-integral-derivative) loop running on the ESP32 SOC. Due to rapid temperature changes, the derivative term can vary significantly between each iteration. To avoid this, the differential term was the moving average of five iterations. This was easy to calculate and reduced oscillations. The integral term was calculated as the sum of the previous iteration’s integral term and the error, multiplied by the time interval between the iterations. PID constants were tuned by the well-known Ziegler–Nichols method. The feedback for PID control was the heater surface temperature, monitored by the thermistor on the heater board. The output of the thermistor was converted by the well-known Steinhart–Hart equation. 

The thermal interface between the heater and heated liquid consisted of two layers, i.e., a ~0.15 mm thick plastic foil laminating the fluidic chip (Greiner MTP sealers, Greiner Holding AG, Austria) and the ~0.02 mm thick solder mask, both negligible compared to the heater area (4 cm^2^) and the microreactor area (~40 mm^2^, including inlet channels). To ensure quick and easy replacement of microfluidic chips, no thermal interface material was used. The heater and the heated microfluidic chip are insulated by the plastic enclosure (this is sufficient due to the microreactor being only ~10% of the total heated area and far smaller than the surrounding enclosure).

### 2.3. Thermal Modelling

The thermal model detailed in this work was designed for stationary thermal analysis and a 3D device geometry with well-defined structural materials and boundary conditions. The model was a simplified 3D representation of the experimental setup. The heater was represented as a temperature boundary condition, given that the steady-state heater surface temperature was known and experimentally verified for each set point. Therefore, the model was based on the heat transfer equation assuming no flow (during amplification, the reaction liquid is stationary). Thermal properties (density, heat conductivity, specific heat capacity) of the structural materials in the model are detailed in Table 2. Boundary conditions and initial values are detailed in Table 3. Ambient temperature for the model was set to a constant 25 °C. 

The model was implemented in COMSOL^®^ Multiphysics version 5.4 using the Heat Transfer interface. The model was solved via the built-in stationary solver of COMSOL on a PC with a Core i7-7700 CPU with 32 GB RAM. The 3D model of the device was imported from Autodesk Inventor and defeatured to decrease mesh complexity: screws and fillets were removed, and the battery and the circuitry were replaced with blocks of the same size. Convective, conductive, and radiative heat losses were taken into account in the model. The model had 36 distinct domains and 1223 boundaries, all of which were included in the calculation. The thermistor in the microreactor was modelled as a Domain Point Probe at the same height as in the experimental model. A tetrahedral mesh was generated in COMSOL using the built-in Finer settings, resulting in 1,301,982 elements of an average element quality (based on the well-known radius ratio method [57]) of 0.63 and element size of 0.3 mm^3^.

### 2.4. Experimental Setup for Thermal Characterization

The experimental setup consisted of the thermostat prototype and a microfluidic chip. The microfluidic chip (3 cm × 2 cm × 1.5 mm) had two inlet and two outlet features and was designed to hold ~50 µL liquid (standard DNA amplification volume [58]). Chip prototypes were 3D printed using a DLP (digital light processing) 3D printer (Envisiontec Perfactory 4, Envisiontec GmbH, Gladbeck, Germany), and the channels of the microfluidic chip were sealed with Greiner MTP sealers (Greiner Holding AG, Austria). For reaction temperature measurements, a bead-type thermistor (P/N GA10K4A1A, TE Connectivity Ltd., Schaffhausen, Switzerland) was embedded in the channel through a hole drilled from above and sealed in with thermoplastic adhesive. The sensor was connected to an Agilent 34410A digital multimeter (Agilent Technologies Inc., Santa Clara, CA, USA) and recorded on a PC via a MATLAB^®^ script. Before experiments, an empty and a water-filled 3 mL syringe were connected to the inlets, and open tubes to the outlets of the microfluidic chip. After filling the reaction chamber with distilled water, the tubes connected to the outlets were clamped shut by IV clamps. Experiment data (timestamped heater temperature, input current and voltage) was recorded by streaming data through the microcontroller’s USB serial interface to PuTTY running on a PC.

## 3. Results and Discussion

### 3.1. Initial Thermal Characterization

Initial characterization was done in an air-conditioned office maintained at 25 °C (± 2 °C) to ensure easy access to the experimental setup. Heater base resistance was measured using the aforementioned Agilent digital multimeter at 3 Ω. Next, thermal characterization was performed on the experimental setup. Three consecutive thermal transients were recorded for 15 min, each time allowing the system to cool back to room temperature afterwards (Figure 2). The goal was to determine the time constant (in this context calculated as the time it took for reaction temperature to enter steady-state with 0.5 °C tolerance) and verify that the steady-state error (SSE) of control was within ±1 °C. For this measurement and the following ones, based on the required SSE, steady-state was defined as starting from being within 1 °C from the set point.

With a set point of 60 °C, the time constant for the heater was 1 min 29 s (± 9 s), for the microreactor it was 5 min 58 s (± 17 s), the steady-state error was 0.35 °C (± 0.06 °C), within the required range. The MCU (microcontroller unit) of the thermostat monitored the input current of the heater and the input voltage from the battery. Using these values, steady-state power was calculated as 0.75 W (± 0.005 W).

### 3.2. Validation of Thermal Simulation Model

To verify the accuracy of the simulation model, infrared images of the experimental setup were recorded (using a Jenoptik VarioCAM 384 HiRes IR camera, Jenoptik AG, Jena, Germany) and compared to the model. Recordings were made at room temperature (25 °C in an air-conditioned office) and the thermostat was set to 60 °C. The model was solved using the Stationary Solver of COMSOL^®^ Multiphysics version 5.4. For a single set of parameters, solution time was between 1–2 min. The same spot was compared on experimental and simulated thermal images (marked by black cross on Figure 3). On the simulated image, the temperature was 56.19 °C, whereas on the experimental image it was 56.04 °C, indicating an estimation error of 0.15 °C on the surface (Figure 3). Please note that the surface of the chip was exposed on these recordings, thus the lower surface temperature than the prescribed setpoint. Comparing the embedded temperature sensors in the microreactor, the absolute error between the simulated and experimental recording was 0.28 °C, within the required SSE range.

### 3.3. Steady-State Thermal Analysis

To assess the heating performance of the thermostat prototype, steady-state thermal analysis was conducted. Experiments were performed in a climate chamber (Vötsch VT 7004, Vötsch Industrietechnik GmbH, Bailingen, Germany) incubated for 30 min before each test at 25 °C to reach the correct ambient temperature and allow the prototype to cool down to ambient temperature. Each experiment involved recording a thermal transient for 15 min with the thermostat set to the defined set point. Tested set points were defined in the 35–65 °C range with 5 °C steps, matching the requirements of 12 distinct isothermal amplification protocols as listed in Table 4. The thermostat prototype was powered from its internal battery. This was achieved by cutting the power wire of the micro-USB connector used for data transfer from the thermostat.

Table 4 summarizes the set points, corresponding steady-state errors, steady-state power consumption values of the heater and time constants related to the reaction temperatures in the microreactor. Power consumption was once again calculated from the input current and voltage drop measured by the thermostat. Each set point was tested twice, and values averaged. The highest time constant was 7 min 8 s for a set point of 65 °C; the lowest was 2 min 39 s for 35 °C. The maximum steady-state power consumption was 0.81 W, the minimum was 0.18 W for the same temperatures. The highest SSE was 0.63 °C for the set point of 40 °C, still within the required boundaries. The lowest SSE of 0.08 °C was recorded at a set point of 55 °C. With applying offsets for the thermistor, the SSE could be reduced close to zero, but the demonstrated prototype was already within the required boundaries for every tested set point. 

Simulated thermal analysis was performed as the next step. However, before the model was used for analysis, it was verified that the estimation error was within the desired SSE range. The model described in Section 2.3 was solved using the Stationary Solver of COMSOL. A Parametric Sweep was set up to include all seven set points. Solution time was 10 min 16 s. Estimation error was calculated as the absolute error between experimentally recorded and simulated thermistor outputs in steady-state. The highest error was 0.53 °C, within the required range. The mean absolute error was 0.25 °C (± 0.16 °C). Table 5 summarizes steady-state temperatures recorded experimentally, compared to simulated values for all tested set points.

In a previous work, we established that ~85 % of the reaction liquid had to be in range for the amplification reaction to conclude successfully [22]. Based on this assumption, at least 42.5 µL of the reaction liquid had to be in the correct range, defined as maximum ±1 °C from the set point. From the simulated thermal model, the reaction liquid volume in range could be estimated. Based on a previous work of ours, we evaluated the volume in range using a logical condition defined in the COMSOL, evaluated for each finite element within the reaction chamber. This can be represented as the following formula: (1)$$\eta =50\cdot \frac{1}{n}\sum_{i=1}^{n}\left(\left(T_{i}>\left(\mathrm{SP}-1\right)\;\left({}^{\circ}C\right)\cap {\;T}_{i}<\left(\mathrm{SP}+1\right)\;\left({}^{\circ}C\right)\right)\in \left\{0;1\right\}\right)$$
where SP denotes the set point, T_i_ denotes temperature values for each finite element, n the total element number and η∈[0;50](μL). 

Table 5 summarizes the estimated reaction volumes in range for each tested set point. The lowest value was 43.32 µL at 60 °C set point, whereas the average was 46.59 µL (± 2.5 µL), both above the required threshold. 

To summarize, we performed steady-state thermal analysis, both experimental and simulated, on the thermostat prototype to verify functionality for various isothermal amplification protocols in 50 µL amplification volume, at 25 °C ambient temperature (Figure 4). The thermostat was tested with set points with 5 °C steps within the 35–65 °C range, corresponding to 12 distinct amplification protocols. We concluded that with the tested parameters, the prototype was capable of regulating reaction temperatures in the desired target ranges with steady-state errors below 0.7 °C. Target temperatures were reached within less than 8 min and maintained with less than 1 W power consumption. Furthermore, we have demonstrated by simulation that >85% of the reaction volume was in range for all tested set points. Therefore, our proposed device design was proven valid for use at 25 °C ambient temperature, for all the listed isothermal amplification protocols. 

### 3.4. Stress Test and Battery Life Estimation

We subjected the thermostat to stress testing. The operating ambient temperature for the thermostat was defined as 20–30 °C, corresponding to the limits of comfortable room temperature. Therefore, we tested whether the prototype could maintain a higher set point at these extremes. As in Section 3.4, testing was performed in the climate chamber, but this time the thermostat was unplugged and running from the internal battery. Three tests were done at both 20 °C and 30 °C ambient temperature, each time recording a transient of 15 min. At 20 °C ambient temperature, the average steady-state reaction temperature was 64.29 °C (± 0.07 °C), heater temperature was 65 °C (± 0.001 °C), with 0.89 W (± 0.06 W) power consumption and SSE of 0.71 °C (± 0.07 °C). The time constant was 8 min 43 s (± 32 s). At 30 °C ambient temperature, the average steady-state reaction temperature was 64.31 °C (± 0.11 °C), heater temperature was 64.99 °C (± 0.001 °C), with 0.77 W (± 0.09 W) power consumption and SSE of 0.69 °C (± 0.11 °C). The time constant was 7 min 20 s (± 8 s). Thus, it was demonstrated that the thermostat was capable of operating within the requirements at the extreme ends of the operating ambient temperature range. 

Finally, we estimated the battery life by running the thermostat at the lower end of the operating ambient range, 20 °C, at the highest allowed set point, 70 °C. Lifetime was calculated from start until the reaction temperature was within ±1 °C of the targeted set point. With a 1200 mAh battery and the defined test conditions, regulation failed at 3 h 15 min 55 s. That is, calculating with an average of 30 min per amplification assay, in the worst case, six isothermal amplifications can be completed with a single charge. This can be easily extended by changing the internal battery for a larger capacity option or connecting the thermostat to a cheap, widely available, portable external battery via micro-USB.

## 4. Conclusions

In this work, we demonstrated a low-cost, wireless smart thermostat for isothermal NAAT in 2 cm × 3 cm LoC cartridges. The prototype thermostat was based on the ESP32 SoC and had Bluetooth connectivity, a 1200 mAh battery, and a built-in OLED display and buttons for local status monitoring and configuration. The prototype heater was a custom PCB with copper tracks acting as the heating circuit, and its output was regulated by an nMOSFET via PWM, based on data from an onboard thermistor. An additional ADC was used to monitor electrical parameters within the system. The prototype enclosure and the LoC cartridge used for the tests were 3D printed. 

The prototype was evaluated experimentally and via finite element modelling, to prove its functionality for various isothermal NAAT protocols. Initial thermal characterization was performed by recording 15-min thermal transients at room temperature with a set point of 60 °C. Meanwhile, the LoC cartridge was filled with water. In this initial test, the time constant was 1.5 min for the heater and 6 min for the microreactor, with an SSE of 0.35 °C. With the same experimental parameters, infrared thermometry was performed and compared to the finite element model for the prototype. The model was capable of estimating experimentally recorded temperatures with an absolute error of 0.28 °C. As the next step, steady-state thermal analysis was performed both via experimental and simulated methods. Set points were defined in the 35–65 °C range characteristic of isothermal NAAT protocols, with 5 °C steps. Each experiment was performed in a climate chamber maintained at 25 °C, and each set point was recorded twice with 15-min thermal transients. The maximum SSE was 0.63 °C, steady-state power 0.81 W and time constant 7 min. Simulated thermal analysis was used to determine how much of the reaction volume was in range. Previously, we established that at least 85% of the volume had to be in range. Of the 50 µL reaction volume in our LoC prototype, at least 43.32 µL was in range for all of the tested set points. The prototype was also tested at the boundaries of its operating ambient temperature range (20–30 °C), with 65 °C set point, to test a worst case. At these parameters, maximum SSE was 0.71 °C, power consumption 0.89 W, time constant 8 min 43 s. The battery life of the prototype was tested at 20 °C ambient with 70 °C set point, to evaluate the worst case. Lifetime was calculated as the time duration until which reaction temperature was regulated within 1 °C of the set point. This resulted in a lifetime of 3 h 15 min 55 s. 

In summary, we demonstrated a smart thermostat prototype generally capable of supporting isothermal NAAT protocols within 35–65 °C range in 2 cm × 3 cm LoC cartridges at 20–30 °C ambient temperatures. The proposed thermostat offers a user-friendly dual user interface (local and remote via Bluetooth), over 3 h battery life on a single charge, compact size (12 cm × 8 cm × 5 cm) and low cost (~50–100 EUR). The key advantages of this prototype are (1) complete portability, (2) remote control and monitoring, (3) good balance between cost and performance. Portability (1) makes field applications possible (e.g., in agriculture, on a farm) by running from the internal battery. Furthermore, the thermostat can be remotely monitored and controlled (2), allowing unattended operation in the field, or controlling several instruments at once. The Bluetooth interface also allows integration and networking with additional instruments (e.g., pumps, sensors, etc.) and automated control from a dedicated computer without any wires that could make field deployment difficult. Finally, the low cost (3) ensures a wider exploitation of isothermal LoC NAAT technology as well as easy integration into low-cost field analysis equipment. As part of a longer plan to deliver low-cost, user-friendly electronics for LoC, the demonstrated thermostat prototype will facilitate large-scale application of isothermal LoC NAAT in research and potentially education. 

## Figures and Tables

**Figure 1 micromachines-10-00437-f001:**
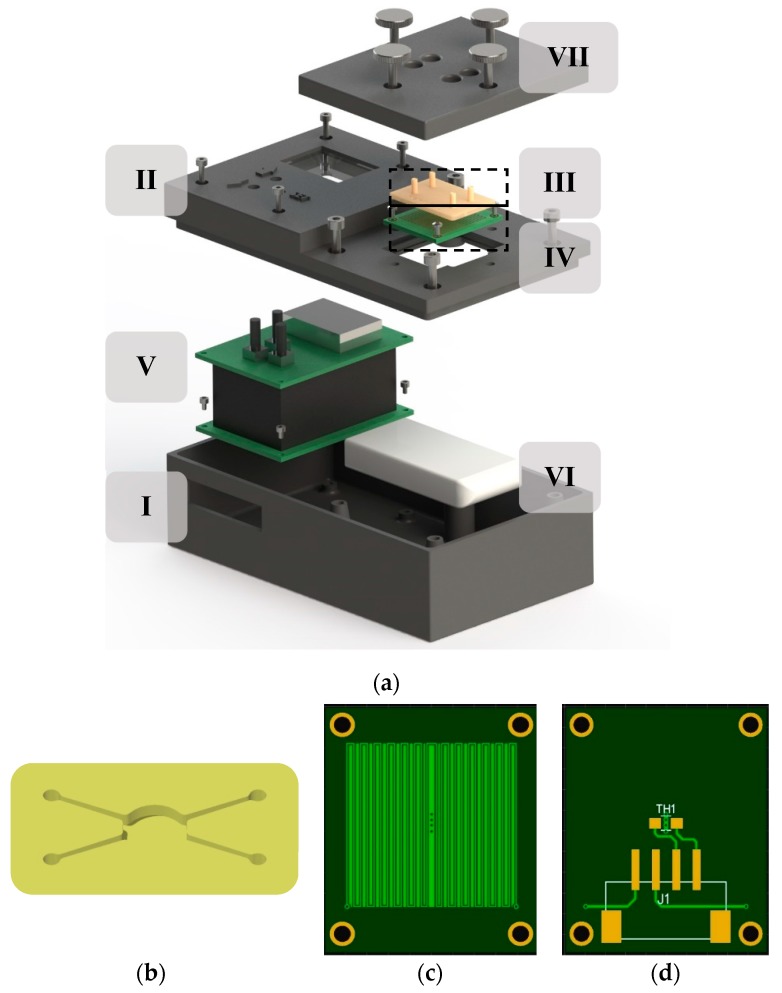
Rendering of the thermostat prototype showing system components (**a**). The prototype was 12 cm × 8 cm × 5 cm and consisted of the following components: Enclosure (I) with user interfaces (II), 2 cm × 3 cm microfluidic chip holding 0.05 mL liquid volume (III), heater (IV), electronics assembly (V), Li-Po battery (VI), and the plastic cover (VII) for clamping down the chip and thermal insulation. The microfluidic reaction chamber (**b**) and the heater (**c**,**d**) are shown separately in magnified view. For easier integration and minimal cost, the heater was implemented on a 2 cm × 3 cm printed circuit board (PCB) board (**c**) with an SMD thermistor on the backside (**d**) for temperature regulation.

**Figure 2 micromachines-10-00437-f002:**
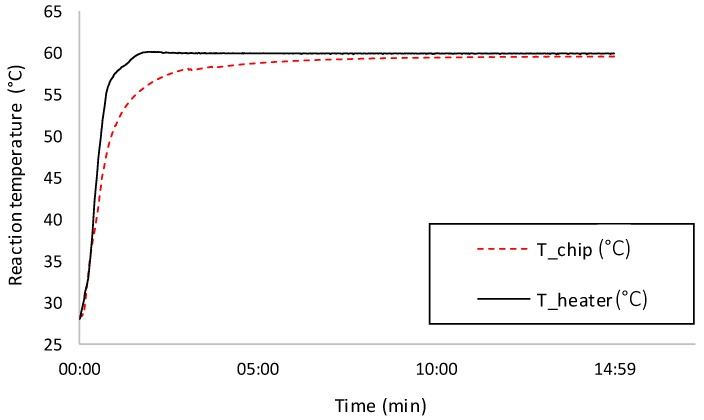
Initial thermal transient characterization of the experimental prototype. With a set point of 60 °C, the time constant for the heater was 1 min 29 s, for the microreactor it was 5 min 58 s, the steady-state error was 0.35 °C. The initial temperature overshoot of the heater was due to integral windup and had no effect on reaction temperature control.

**Figure 3 micromachines-10-00437-f003:**
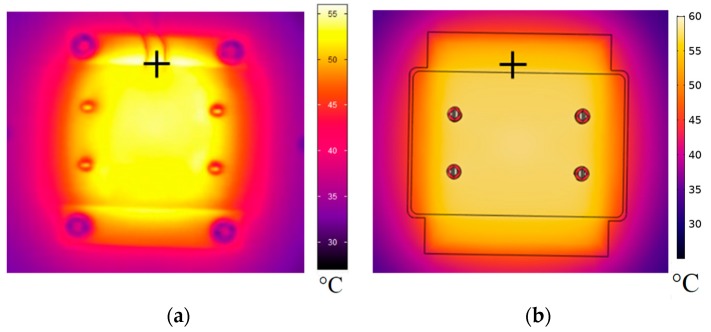
Steady-state surface temperature distribution on infrared image of experimental setup (**a**) and simulated thermal model (**b**). The thermal model was defeatured (screws, filleting etc. were removed) to decrease computational complexity for the model. Directly comparing temperatures in the same spot (marked by a black cross) on experimental and simulated thermal images indicated an absolute error of ~0.2 °C.

**Figure 4 micromachines-10-00437-f004:**
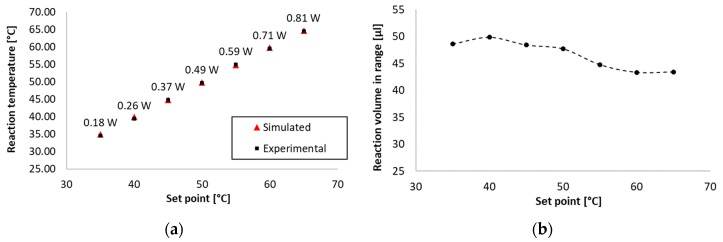
Experimental and simulated steady-state temperatures at various set points (**a**) as well as estimated reaction volume in range (**b**) corresponding to them. Power consumption is shown above temperature data points. Temperature data points were recorded in the 35–65 °C range with 5 °C steps, at 25 °C ambient temperature, corresponding to 12 isothermal amplification protocols performed in a regular office/laboratory environment. The prototype was demonstrated capable of performing these protocols with respect to reaching and maintaining target temperatures as well as holding above 85% of the reaction liquid in the required temperature range.

**Table 1 micromachines-10-00437-t001:** Comparison of temperature control solutions for Lab-on-a-Chip (LoC) nucleic acid amplification tests (NAAT) devices reported in the literature.

Solution	Heating Element	Control	LoC Interface & Insulation	Any Isothermal NAAT?	End-User Cost (Estimated) (€)	Source
Hot plate	Resistive	PID	No	Yes	>500	[35,45]
External (products only)	Peltier/ resistive	PID	Examples exist	Yes	Peltier > 2500 Resistive > 500	[26,36,44]
Integrated	Resistive/ µPeltier	PID or PTCR effect	Integrated to LoC	No	>500	[20,21,22,23,41]
Chemical heating	Exothermic reaction	PCM	No	No	<10	[27,28,29,30]
This work	Resistive	PID	Yes	Yes	100	-

**Table 2 micromachines-10-00437-t002:** Summary of material properties used in the model.

Material	Density (kg/m^3^)	Thermal Conductivity (W/mK)	Specific Heat Capacity (J/(kg·K))
3D printed plastic	1470	0.18	1190
Copper	8960	400	385
FR4	1900	0.143	1369
Air	1225	0.024	1000
Water	1000	0.6	4184

**Table 3 micromachines-10-00437-t003:** Summary of boundary conditions and initial parameter values used in the model.

Boundary Condition	Boundary	Initial Value (If Applicable)
Ambient temperature	External boundaries	25 °C
Ambient pressure (absolute)	External boundaries	1 atm
Heater	Heater track surface	As per set point
Convective heat loss	External boundaries	Not applicable
Electrical insulation	Heater boundaries except contacts	Not applicable
Radiative heat loss	External boundaries	Not applicable

**Table 4 micromachines-10-00437-t004:** Summary of experimental thermal analysis including time constants, power consumption and steady-state error for various nucleic acid amplification test (NAAT) protocols [31].

NAAT Protocol	Target Range (°C)	Set Point (°C)	Steady-State Error (SSE) (°C)	Steady-State Power (W)	Time Constant
RAM	35	35	0.48	0.18	2 min 39 s
RPA, BAD AMP	37–42	40	0.63	0.26	2 min 47 s
SPIA	45–50	45	0.17	0.37	5 min 41 s
	45–50	50	0.37	0.49	6 min 29 s
EXPAR, NEAR	55–55	55	0.08	0.59	6 min 56 s
TMA, ICA, PG-RCA	60	60	0.46	0.71	7 min 8 s
LAMP, CPA, NEMA	65	65	0.46	0.81	7 min 30 s

**Table 5 micromachines-10-00437-t005:** Summary of simulated thermal analysis for various nucleic acid amplification test (NAAT) protocols [31].

NAAT Protocol	Target Range (°C)	Set Point (°C)	Experimental Steady-State (°C)	Simulated Steady-State (°C)	Volume in Range (µL)
RAM	35	35	34.52	34.94	48.62
RPA, BAD AMP	37–42	40	39.37	39.90	49.89
SPIA	45–50	45	44.83	44.87	48.43
	45–50	50	49.63	49.83	47.72
EXPAR, NEAR	55–55	55	54.92	54.80	44.74
TMA, ICA, PG-RCA	60	60	59.54	59.77	43.32
LAMP, CPA, NEMA	65	65	64.54	64.73	43.408

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
