# Peer review of "Development of a Low-Cost, Wireless Smart Thermostat for Isothermal DNA Amplification in Lab-On-A-Chip Devices"

_micromachines, 2019, doi:10.3390/mi10070437_

Round 1
Reviewer 1 Report
In this manuscript, the authors reported a low-cost heating device for performing isothermal DNA amplification for point-of-care diagnostics. The device is controlled by an external wireless thermostat with a good balance between cost and performance. The design of the device is technically sound, and both numerical simulation and experimental tests were performed. It is a little pity that no actual amplification assays have been done on this heating device though. Nevertheless, the manuscript may be still interesting to the readers of Micromachines from the device design and prototyping point of view.
Minor comments include:
1. The benefit of the wireless function of the thermostat is not clear to this reviewer. It seems that all previous close-loop feedback thermostats work well. Why do we need the wireless? The authors may need to add more discussion to make the point clearer.
2. Please define all acronyms at their first use, e.g. PID at line 46 on page 2. Please also define the term “time constant”. It is not clear to this reviewer the meaning of this term and how it was calculated.
3. What is the height of the microfluidic chamber? Is there a way to measure the 3D temperature distribution of the chamber? How to further increase the liquid volume in the desired temperature range? Maybe make the chamber flatter?
4. The reference format needs to be thoroughly corrected.
Author Response
Dear Reviewer #1,
Thank you for your comments and suggestions. I will copy your remarks below and respond under them in red, as well as list the changes made to the manuscript.
---
1. The benefit of the wireless function of the thermostat is not clear to this reviewer. It seems that all previous close-loop feedback thermostats work well. Why do we need the wireless? The authors may need to add more discussion to make the point clearer.
Yes, thank you for pointing this out, the conclusion was extended to include an explanation. Portability is the most important aspect of this design. Lab-on-a-Chip instrumentation is getting stronger recently in benchtop devices (e.g. Dolomite has a dedicated chip temperature controller), but portability is still a largely uncovered issue. So wireless goes two ways: portable in the sense of power supply (no wires) and remote control. The former is to ensure field applications are possible (e.g. at a farm or next to a food processing line). The latter is mostly so that the device can run unattended, even in the field.
2. Please define all acronyms at their first use, e.g. PID at line 46 on page 2. Please also define the term “time constant”. It is not clear to this reviewer the meaning of this term and how it was calculated.
Defined calculation of time constant (ln 207) and added definitions in text. If any acronyms remain undefined by oversight, during the final edit they will be added.
3. What is the height of the microfluidic chamber? Is there a way to measure the 3D temperature distribution of the chamber? How to further increase the liquid volume in the desired temperature range? Maybe make the chamber flatter?
In the tested chip demonstrator, the height was 1.2 mm. We can’t measure the 3D temperature distribution inside the chamber, this is why we use finite element modelling to intra/extrapolate from experimental temperature data and well-established library parameters. It is possible to redesign the reaction chamber to suit the application better, as long as the overall chip size does not exceed 2 cm x 3 cm x 1.5 mm (the height can be increased to 2 mm if need be). If the microfluidic channel geometry is changed, the thermal verification process remains the same as was discussed in this manuscript.
4. The reference format needs to be thoroughly corrected.
The MDPI reference format built into Mendeley was used to generate references. If changes are necessary, the editors will inform us during the next step of the review process and changes will be made.
---
One additional remark: We would have liked to test the device prototype with DNA amplification, but we had neither the time nor the resources to do so for this project. We are looking for partners to do so and it is definitely something we intend to do later.
Kind regards (on behalf of the whole team of researchers),
Dr. Tamas Pardy
Researcher
Thomas Johann Seebeck Department of Electronics
Tallinn University of Technology
Reviewer 2 Report
The authors have presented a wireless thermostat with Bluetooth 325 connectivity in 2 cm x 3 cm LoC cartridges.
The contents of the paper are well organized. I would recommend the manuscript for publication in Micromachines, following minor revision, as stated below:
1. Figure 3b: State the unit of temperature in the color bar.
2. Figure 4a: It is difficult to observe the simulated data points. Please decrease the size of the black markers to make them visible.
3. Is the microfluidic chip one-off use chip? If yes, how does the thermal panel interact with the microfluidic flow? If no, how many tests can it handle?
4. What is the range of remote operation possible via Bluetooth. The authors have not shown any of such results.
Author Response
Dear Reviewer #2,
Thank you for your comments and suggestions. I will copy your remarks below in bold and respond under them in red, as well as list the changes made to the manuscript.
1. Figure 3b: State the unit of temperature in the color bar.
Temperature unit is now clearly indicated at the bottom of each color bar.
2. Figure 4a: It is difficult to observe the simulated data points. Please decrease the size of the black markers to make them visible.
The size of black markers was decreased. However, the numerical data is also available in Table 4 for readers.
3. Is the microfluidic chip one-off use chip? If yes, how does the thermal panel interact with the microfluidic flow? If no, how many tests can it handle?
We envisioned a single-use chip, made from cold-laminated 3D printed plastic. In a commercial setting, chips could be injection-moulded and laminated to make them cheap. In this study we did not focus on flow characterisation, but from an another study using similar chips we concluded that pressure-induced flow may push minimal amounts of the reaction liquid away from the heated zone, but not enough to be detrimental to the reaction yield. This effect can be countered by chip geometry optimization.
4. What is the range of remote operation possible via Bluetooth. The authors have not shown any of such results.
The range of the Bluetooth transmission for the ESP32 in our device was standard, up to 10 m in an office environment. This was confirmed by others (https://www.esp32.com/viewtopic.php?t=6959). Added a sentence in line 105 to make this clear to readers.
Kind regards (on behalf of the whole team of researchers),
Dr. Tamas Pardy
Researcher
Thomas Johann Seebeck Department of Electronics
Tallinn University of Technology